# Hepatitis B virus compartmentalization and single-cell differentiation in hepatocellular carcinoma

Frank Jühling[1],*, Antonio Saviano[1,2],*, Clara Ponsolles[1], Laura Heydmann[1], Emilie Crouchet[1], Sarah C Durand[1], Houssein El Saghire[1], Emanuele Felli[1,2], Véronique Lindner[3], Patrick Pessaux[1,2], Nathalie Pochet[4,5], Catherine Schuster[1,2], Eloi R Verrier[1], Thomas F Baumert[1,2,6]

Chronic hepatitis B virus (HBV) infection is a major cause of hepatocellular carcinoma (HCC) world-wide. The molecular mechanisms of viral hepatocarcinogenesis are still partially understood. Here, we applied two complementary single-cell RNA-sequencing protocols to investigate HBV–HCC host cell interactions at the single cell level of patient-derived HCC. Computational analyses revealed a marked HCC heterogeneity with a robust and significant correlation between HBV reads and cancer cell differentiation. Viral reads significantly correlated with the expression of HBV-dependency factors such as *HLF* in different tumor compartments. Analyses of virus-induced host responses identified previously undiscovered pathways mediating viral carcinogenesis, such as E2F- and MYC targets as well as adipogenesis. Mapping of fused HBV–host cell transcripts allowed the characterization of integration sites in individual cancer cells. Collectively, single-cell RNA-Seq unravels heterogeneity and compartmentalization of both, virus and cancer identifying new candidate pathways for viral hepatocarcinogenesis. The perturbation of pro-carcinogenic gene expression even at low HBV levels highlights the need of HBV cure to eliminate HCC risk.

## Introduction

Hepatitis B virus (HBV) infection is a major cause of chronic liver disease and hepatocellular carcinoma (HCC) (El-Serag, 2012). An estimated 2 billion people have evidence of exposure to HBV and more than 250 million people are chronically infected with HBV worldwide. HBV-infected patients have an ~100-fold increased risk for HCC compared to uninfected patients (Perz et al, 2006). HCC is the second leading and fastest rising cause of cancer death worldwide (Bray et al, 2018). Each year, close to 600,000 people are newly diagnosed with HCC. The future significance and impact of the disease is not only illustrated by its rising incidence over the last two decades, but also by its unchanged high mortality (Flores & Marrero, 2014). Thus, the burden of established, incurable HBV-induced liver disease represents a major challenge to public health, and efficient treatment strategies to cure chronic hepatitis B (CHB) are urgently needed (Flores & Marrero, 2014). The pathogenesis of HBV-induced HCC is multifactorial. CHB induces HCC through direct and indirect mechanisms (reviewed in Levrero and Zucman-Rossi [2016]). First, the viral DNA has been shown to integrate into the host genome inducing both genomic instability and direct insertional mutagenesis of diverse cancer-related genes (Levrero & Zucman-Rossi, 2016). Compared with tumors associated with other risk factors, HBV-related tumors have a higher rate of chromosomal alterations including p53 inactivation by mutations. Moreover, epigenetic changes targeting the expression of tumor suppressor genes (TSGs) have been shown to occur early in the development of HCC (Levrero & Zucman-Rossi, 2016). Second, HBV proteins such as the viral regulatory protein HBx or altered versions of the preS/S envelope proteins have been shown to modulate cell transcription, resulting in alteration in host cell proliferation and sensitizing the hepatocytes to carcinogenic factors (Levrero & Zucman-Rossi, 2016). HBV-related HCCs can also arise in non-cirrhotic livers, supporting the notion that HBV plays a direct role in liver transformation by triggering both common and etiology-specific oncogenic pathways in addition to stimulating the host immune response and driving liver chronic necro-inflammation (Levrero & Zucman-Rossi, 2016). The question of HBV replication within tumors has been studied for more than 50 yr (Aden et al, 1979). Interestingly, it has been recently suggested that HBV replicates in highly differentiated tumors (Halgand et al, 2018; Rivière et al, 2019), but little is known about viral–host interactions and HBV contribution to HCC progression once HCC is established. Single-cell RNA-Seq is a high-resolution technique allowing transcriptome-wide

[1]Université de Strasbourg, Inserm, Institut de Recherche sur Les Maladies Virales et Hépatiques UMR_S1110, Strasbourg, France   [2]Institut Hospitalo-Universitaire, Pôle Hépato-digestif, Nouvel Hôpital Civil, Strasbourg, France   [3]Hôpitaux Universitaires de Strasbourg, Département de Pathologie, Strasbourg, France   [4]Ann Romney Center for Neurologic Diseases, Department of Neurology, Brigham and Women's Hospital, Harvard Medical School, Boston, MA, USA   [5]Cell Circuits Program, Broad Institute of MIT and Harvard, Cambridge, MA, USA   [6]Institut Universitaire de France (IUF), Paris, France

Correspondence: e.verrier@unistra.fr; thomas.baumert@unistra.fr
Eloi R Verrier and Thomas F Baumert contributed equally as senior authors
*Frank Jühling and Antonio Saviano contributed equally to this work

analyses of individual cells, and represents a precious tool to study heterogeneous tissues including cancer (Sandberg, 2014). Tumors are characterized by multiple neoplastic sub-clones as well as non-neoplastic cells constituting the tumor microenvironment. Moreover, single-cell RNA-Seq enables to distinguish cells with different levels of HBV infection, cells exposed to the virus but not infected as well as noninfected cells. Here, we applied two different single-cell RNA-sequencing protocols to investigate HBV–HCC host cell interactions at the single cell level of patient-derived HCC.

# Results

## Clinical characteristics of HCC patients with chronic HBV infection included in the study

To study HBV–HCC interactions we included two HCC patients with CHB who are representative for the patient population. Both were followed at a tertiary referral center. The first patient (P1) was an untreated HBV-inactive carrier and the second patient (P2) was a patient having received antiviral treatment. Patient P1 was 61-yr-old with normal liver function tests. HBV load, which predicts the risk of disease progression in patients with chronic HBV infections (Chen et al, 2007), was low at 1.85 $\log_{10}$ IU/ml. Additional risk factors for HCC were type-2 diabetes and a family history of HCC and hemochromatosis. Both HCC and HBV infection were diagnosed after the patient presented to the emergency room for hemoperitoneum due to HCC rupture. Serological tests for hepatitis C virus (HCV), hepatitis D virus, and HIV were negative; levels of alpha-fetoprotein (AFP) and CA 19-9 were within the normal range. Hemochromatosis was ruled out by clinical chemistry and *HFE* gene C282Y/H63D/S65C analysis (Table S1). A left hepatectomy was performed and the histopathological examination of the resected tissue revealed a well to moderately differentiated HCC displaying both a trabecular and a pseudoglandular pattern classified as G1-G2 according to the Edmondson-Steiner classification (Edmondson & Steiner, 1954) (Fig S1A–F). The surrounding liver tissue showed portal fibrosis with some incomplete septa (METAVIR F2) (Bedossa & Poynard, 1996) and mild macrovesicular steatosis (Fig S2A–D) without evidence of lobular inflammation nor hepatocyte ballooning. After surgical recovery, an antiviral therapy with entecavir was started and 18 mo after treatment start, HBV-DNA were undetectable and HBsAg was lost. No HCC recurrence was detected during a 3-yr follow-up. The second patient (P2) was a 28-yr-old man who had an incidental diagnosis of single liver tumor of 6 cm and a chronic HBV infection. At the time of the diagnosis, transaminases were elevated at 1–2 × of upper normal limit and HBV DNA level was 3.77 $\log_{10}$ UI/ml. Serological tests for HCV, hepatitis D virus and HIV infection were negative. The patient refused any invasive procedures or treatments for the liver tumor and an antiviral HBV treatment consisting of tenofovir 245 mg/d was started. After 1 yr, the patient presented to our outpatient clinic for abdominal pain and new imaging showed a 12-cm liver tumor in the right lobe with vascular invasion of right hepatic vein, inferior vena cava, and portal vein. Right extended hepatectomy and thrombectomy under extracorporeal circulation and in-situ hypothermic perfusion with caval

reconstruction by a bovine pericardial patch were performed. The histological analysis showed a moderately differentiated HCC displaying thick trabeculae classified G2 according to the Edmondson-Steiner classification (Edmondson & Steiner, 1954). At the time of surgery, HBV-DNA was 2.17 $\log_{10}$ UI/ml under antiviral treatment. Histological analysis of adjacent non-tumor liver tissue revealed portal fibrosis with some incomplete septa (METAVIR F2) (Bedossa & Poynard, 1996). After surgery, the patient developed an early tumor recurrence with liver and pulmonary metastasis. He was treated with sorafenib without response and died after 6 mo from the surgery. Patients' clinical characteristics are presented in Table S1.

## Single-cell analysis of well- and moderately differentiated HBV-associated HCC identifies liver carcinogenetic pathways and cancer differentiation trajectories

To investigate HBV–host interactions in HCC at a single-cell level, we isolated single cells from the resected HCC tissues from these two patients, both presenting active HBV replication at time of the surgery. No necrosis was observed from liver resection. In a first approach single-cell gene expression was analyzed using mCEL-Seq2 (Muraro et al, 2016). Nine hundred thirty-eight cells from the two patients passed the filtering and quality control and were clustered and analyzed independently from HBV-RNA levels (Fig 1A and B). A summary of the number of sequenced cells is presented in Table S2. To illustrate differences and similarities among single cell between the two patients, we artificially plotted single-cell gene expressions on T-distributed stochastic neighbor embedding (*t*-SNE) maps indicating cell similarities as commonly performed in (liver) single-cell studies (MacParland et al, 2018; Aizarani et al, 2019; Losic et al, 2020). Applying an unsupervised clustering resulted in 22 different cell clusters highlighting a pronounced cancer cell heterogeneity (Fig 1B). Clusters of single cells were characterized using known markers as previously described (Aizarani et al, 2019) (Fig 1C). Interestingly, the comparison of patient origin (Fig 1A), unbiased clustering (Fig 1B), and marker gene identification (Fig 1C) showed that cells composing the tumor microenvironment tend to overlap in the *t*-SNE, whereas HCC epithelial cells are clearly separated. This suggests that for these patients, the tumor microenvironment appears to be less heterogeneous than HCC epithelial cells although both tumors are HBV-related. Whereas P1-derived cells expressed high levels of hepatocyte-specific genes such as *ALB*, P2-derived cells highly express classical HCC markers such as *GPC3* (Ho & Kim, 2011) (Fig 1D). In line with this, single-cell gene set enrichments revealed enrichment in HCC P1 for Boyault's HCC subclass G5-G6, which is strongly related to beta-catenin mutations, higher tumor differentiation, and better patients' survival, whereas Boyault's subclass G1, representing less differentiated tumors with chromosome instability, is enriched in HCC P2 (Boyault et al, 2007) (Fig S3A and B). Even though both patients had similar serum HBV-DNA levels, HBV-RNAs were found to be much higher expressed in P1 than in P2 tumor cells (Fig 1D). To investigate tumor differentiation, we compared gene expression in single HCC cells from both tumors, with single-cell transcriptomic data of 200 healthy, randomly selected hepatocytes from the recently published human liver cell

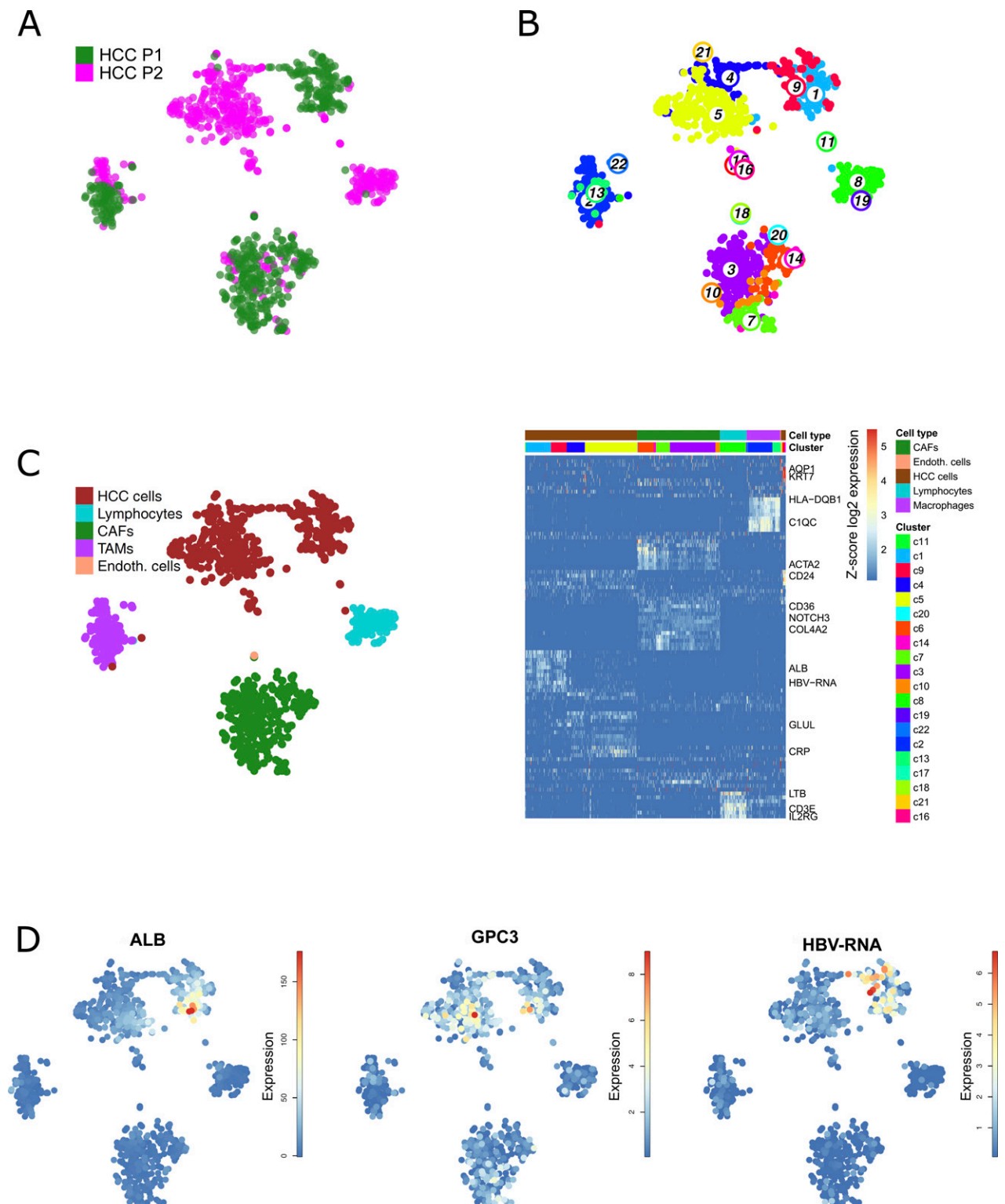

**Figure 1.   Single-cell RNA sequencing of well- and moderately differentiated HCC.**
**(A)** *t*-SNE map of well-differentiated (HCC P1) and moderately differentiated HCC (HCC P2). **(B)** Unsupervised k-medoids clustering of all single cells resulted in 22 clusters. **(C)** Cell-type annotation using known markers according to Aizarani et al (2019). **(B)** A t-SNE map (left) shows main cell type annotations as identified according to marker gene expressions as highlighted in the heat map (right). Colors indicating cell clusters refer to (B). **(D)** Expression *t*-SNE of albumin (*ALB*), tumor marker glypican-3 (*GPC3*), and HBV-RNA levels.

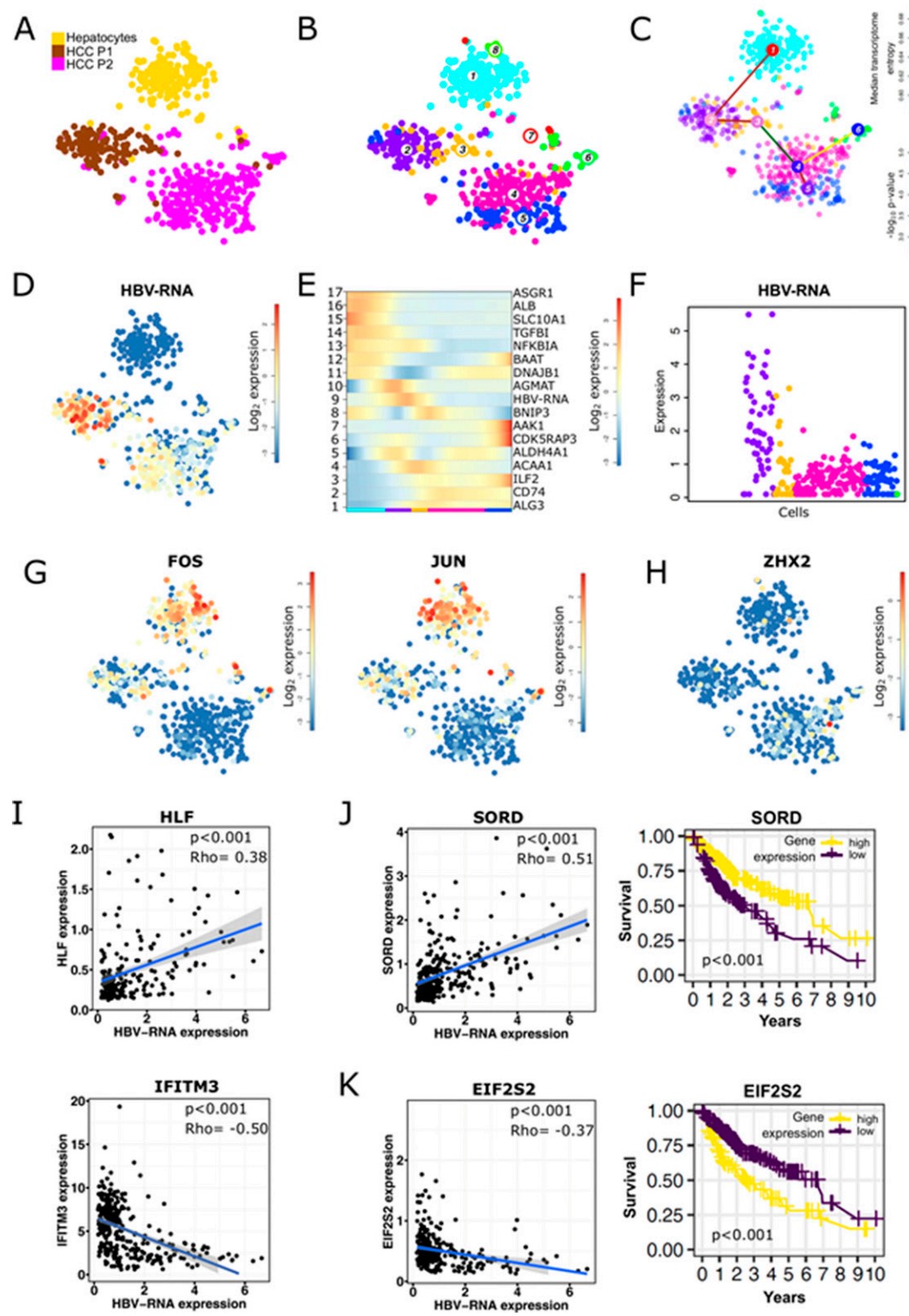

**Figure 2. Intertumor heterogeneity of HBV-RNA expression and virus-host factor interaction analysis.**
**(A, B, C, D, F)** HBV-RNA levels are linked to the differentiation state of the cells. **(A)** *t*-SNE map of HCC P1, HCC P2, and healthy hepatocytes from Aizarani et al (2019).
**(B)** Unsupervised and HBV-independent k-medoids clustering of hepatocytes and cancer cells resulted in eight clusters. **(C)** Differentiation lineage reconstruction connecting hepatocytes (cluster 1) progressively with less differentiated cancer cells (cluster 2–6). **(D)** Expression *t*-SNE of HBV-RNA. **(E)** Self-organizing map of pseudo-temporal expression profiles along the differentiation branch; one representative gene from each module is shown. **(F)** Single-cell HBV-RNA expression along the differentiation lineage. Colors indicating cell clusters refer to (B), and only cells originating from HBV-infected tumors are shown. **(G, H)** Expression *t*-SNE of HBV–host factors. **(G)** Expression *t*-SNE *FOS* and *JUN*, known HBV replication enhancers, show higher expression in hepatocytes and HCC P1. **(H)** Expression *t*-SNE *ZHX2*, a known HBV

atlas (Aizarani et al, 2019) as a reference (Fig 2A). Unbiased, HBV-RNA–independent cell clustering (Fig 2B) followed by a lineage reconstruction was performed (Aizarani et al, 2019). Although this approach does not imply an HCC monophyletic evolution from healthy hepatocytes to P2 HCC cells, it allowed to infer unbiased cell differentiation. The analysis revealed a lineage tree consisting of a single differentiation branch connecting clusters 1 (healthy hepatocytes), 2 and 3 (HCC P1), and 4 (HCC P2), and then bifurcating into HCC P2 clusters 5 and 6 (Fig 2C). On the inter-tumor level, our lineage analysis confirms unbiasedly the histological grading of both HCCs on the single-cell level (Fig 2A–C). Clusters 7 and 8 were not included in the lineage tree, lacking strong links with any other cluster. We then evaluated pseudo-temporal expression profiles along the differentiation branch to identify differentially expressed between the different clusters. Self-organizing maps (SOM) of the pseudo-temporal expression profiles were calculated, grouping genes of similar expression profiles into joint modules (Fig 2E). This allowed to identify similar gene expression profiles along the cancer differentiation trajectory, which shows (I) a loss of hepatocyte differentiation markers such as *ASGR1*, *SLC10A1*, and *ALB* (Uhlén et al, 2015) early in P1 tumor, (II) an early expression of inflammatory genes, for example, *TGFBI* and *NFKBIA* suggesting their role in the carcinogenetic process, and (III) a gain of cancer-associated marker gene expression, for example, *ILF2* and *CDK5RAP3* late in the P2 tumor. The different expression profiles along the trajectory from cluster 1 to cluster 6 highlights their critical roles during early and late phases of HCC development (Mak et al, 2011; Cheng et al, 2016) (Fig 2E). Taken together, our results dissect the intertumoral heterogeneity associated with HCC differentiation at the single-cell level, reconstruct HCC differentiation trajectories, and identify genes differentially expressed in early and late phases of HCC development.

### HBV-RNA levels positively correlate with tumor differentiation markers and HBV-related host factors at the inter-tumor level

Using bulk RNA-Seq analyses of HCCs, Losic and colleagues have recently observed that HBV RNA expression levels vary among HCCs from different patients and suggested that this variation between patients is not dependent on viral antigenicity nor antiviral immune responses (Losic et al, 2020). However, because of the absence of in-depth single-cell RNA-seq analyses of HBV in individual cancer cells, the heterogeneity of the virus within a patient HCC and its dependency of the cancer cell phenotype remains unknown. To address this important question, we analyzed HBV reads in HCC single cells enabling to unravel the relationship between HBV expression, tumor phenotype and host gene expression in individual tumor cells. Expression *t*-SNE maps clearly show that, despite comparable viral loads in both patients' serum, higher levels of HBV reads were detected in the HCC P1 compared to the HCC P2, and higher levels of HBV reads were observed in the most differentiated cluster (cluster 2) within the HCC P1 (Fig 2B–D). HBV read levels along the differentiation trajectory confirm a strong positive

association between cell differentiation and HBV reads (module 9 in Fig 2E and F). Confirming this hypothesis, gene set enrichment analysis (GSEA) from module 9 using the GSEA online "Investigate Gene Sets" tool revealed significant enrichments of 16 genes (False discovery rate [FDR] = $1.43 \times 10^{-12}$) associated with liver specific genes (Hsiao et al, 2001), and of 13 (FDR = $3.28 \times 10^{-9}$) genes associated with a more differentiated tumor phenotype according the Hoshida "subtype S3" signature (Hoshida et al, 2009) (see Table S3). Both sets include alcohol dehydrogenase-encoding *ADH6*, highly expressed in hepatocytes (Uhlén et al, 2015) as their top hit. Interestingly, along with HBV-RNA in module 9 were found *AKR1C1* and *FABP1* whose expressions are known to be up-regulated by HBx (Li et al, 2013; Wu et al, 2016). We next asked the question whether HBV expression in individual HCC cancer cells correlates with the expression of host-dependency factors of the viral life cycle such as HBV enhancers. Indeed, *KRT8* and *HLF*, encoding factors known to enhance HBV replication (Zhong et al, 2014; Turton et al, 2020), were also found to be elevated in module 9 together with elevated HBV reads. Moreover, higher HBV expression levels in HCC P1 and P2 cells were associated with higher expression of *FOS* and *JUN*, both encoding the activator protein-1, known to enhance HBV transcription (Turton et al, 2020) (Fig 2G). On the other hand, the expression of *ZHX2*, encoding the zinc finger and homeoboxes 2 protein, a liver-enriched tumor suppressor known to inhibit HBV transcription (Turton et al, 2020) was found enriched in P2 cells (Fig 2H). In contrast, we did not observe an association with HBV entry factor sodium taurocholate cotransporting polypeptide (NTCP, encoded by *SLC10A1*). A comprehensive analysis of the expression of HBV-host dependency factors and HBV reads is provided in Table S4. Interestingly, although HBV expression was higher in HCC P1 cells, the expression of several HBV host factors, positively associated with HBV replication, were found to be up-regulated in HCC P2. This may reflect their involvement in tumor proliferation and progression. In line with this result, we observed a significant positive correlation between HBV-RNA levels and the expression of *HLF*, a well described liver-specific transcription factor crucial for HBV replication (Turton et al, 2020). On the other hand, HBV expression correlated negatively to *IFITM3* expression, a classical antiviral interferon stimulated gene (Smith et al, 2014) (Fig 2I) suggesting an interplay with the virus and cancer cell innate immune responses. Collectively, HBV expression correlated with the level of cellular differentiation, known to be important for HBV replication (Gripon et al, 2002; Quasdorff et al, 2008). Because viral RNA expression correlated with tumor differentiation, we further characterized the phenotype of the HBV expressing cells using the survival data gene sets from TGCA. We found that *SORD*, encoding sorbitol dehydrogenase, and highly expressed in differentiated hepatocytes correlated with HBV expression (Uhlén et al, 2015) (Fig 2J). On the other hand, *EIF2S2*, encoding the eukaryotic translation initiation factor 2 subunit β associated with an unfavorable outcome in less differentiated HCC, inversely correlated with HBV-RNA

replication inhibitor, which is higher expressed in HCC P2. **(I)** Single-cell correlation analysis between HBV-RNA and *HLF*, a HBV-host factor enhancing viral replication, and *IFITM3*, an antiviral protein, shown for P1 and P2 in cancer cells; *P*-values for spearman correlation. **(J)** HBV-RNA correlates positively in P1 and P2 cancer cells with *SORD*, encoding for sorbitol dehydrogenase, a novel good prognosis marker in liver cancer patients. **(K)** HBV-RNA correlates negatively with *EIF2S2*, a translation initiation factor and a novel poor prognosis marker in liver cancer patients. Kaplan–Meier curves of survival data of liver cancer patients from TCGA data.

(Uhlén et al, 2015) (Fig 2K). This correlation analysis using prognostic markers confirms the lineage analysis prediction and confirms that less differentiated cancers with poor prognosis express lower HBV-RNA. Recently, it was shown that HBV replicates in HCC (Rivière et al, 2019), especially in well-differentiated tumors (Halgand et al, 2018). To assess whether HBV reads are associated to HBV replication, we quantified both HBV covalently closed circular DNA (cccDNA) and HBV pgRNA in both tumor samples. Interestingly, cccDNA and pgRNA were detected in P1 tumor tissue, confirming an active replication of the virus within the highly differentiated tumor (Fig S4A and B). Undetectable cccDNA and pgRNA levels in P2 may be due to technical issues (repeated freezing and thawing), alternatively the HBV reads in P2 correspond to transcripts expressed from integrated HBV DNA into the host genome. Our data thus demonstrate, for the first time at single-cell level, that HBV replication and expression appears to be dependent of differentiation of cancer cells and that this dependency is most likely a result of dysregulation of the expression of specific host factors that are crucial for the HBV life cycle.

### Fine resolution of intratumor and viral heterogeneity of HCC using Smart-Seq2 reveals novel pathways in HBV–cancer cell interactions

To fine-map the heterogeneity of virus and cancer within the same tumor, we sequenced an additional 66 single cancer epithelial cells from the P1 HCC using the Smart-Seq2 protocol that allows an ultra-high coverage of full-length RNA transcripts for a limited number of cells. Smart-Seq2 analyses of primary human hepatocytes isolated (PHH) from a healthy donor were used as a reference for hepatocytes (Fig 3A). PHH and HCC cells were integrated and clustered independently from HBV expression (Picelli et al, 2014; Herman et al, 2018) (Fig 3B). Clustering based on SmartSeq2 categorized PHH into an independent cluster (cluster 1), and separated HCC P1 cells into four distinct clusters (Fig 3B). Notably, whereas clusters 2 and 3 highly express liver specific markers *ALB* and *APOC1*, cluster 4 exhibits high levels of *DDX5*, which plays an important role in HCC proliferation (Xue et al, 2018) (Fig 3B). Single-cell gene set variation analysis revealed that cells from clusters 2/3 and 4 exhibit very distinct features overall. Notably, genes associated with HCC differentiation as Hoshida subclass S3 (Hoshida et al, 2009) are up-regulated in clusters 2/3 (Fig 3C). Hoshida subclass S3 HCC are associated to hepatocyte differentiation and are characterized by high expression of p53 and p21 target gene sets as well as hepatocyte function-related genes (Hoshida et al, 2009). By contrast, cluster 4 is consistently enriched for genes involved in epithelial cell proliferation as determined by the Gene Ontology gene set (0050673) and for genes highly expressed in HCC cells with hepatic stem cell properties, including Wnt/beta-catenin signaling components triggering HCC development (Yamashita et al, 2009). Finally, the expression of genes belonging to the DNA repair machinery as defined by the Hallmark gene set collection (Liberzon et al, 2015) is down-regulated in cluster 4. This is consistent with previous findings of higher chromosomal instability in subclasses with high proliferation (Chiang et al, 2008) (Fig 3C). Moreover, growth factor activity is higher and HCC-specific growth factors *VEGFA* and *EGFR* (Fig S5A and B) are higher expressed in cluster 4 as compared with the others. A similar heterogeneity was

observed when analyzing the expression of gene sets associated with patients' prognosis with HCC. Intersecting differentially expressed cluster marker genes at *P*-value <0.01 with prognostic genes in liver cancer (identified in patients' data from The Cancer Genome Atlas [TCGA] and listed in the Human Protein Atlas) resulted in 425 hits listed as "prognostic, unfavorable" and corresponding to poor patient's prognosis. This poor prognostic liver cancer genes were found in large part in cluster 4 (274/425, 64.5%) and in cluster 5 (123/425, 28.9%). Top 25 marker genes prognostic for poor patients' outcome for HCC clusters 2–5 are presented in Table S5 and examples of such poor prognosis marker genes in cluster 4 are shown in Fig 3D. Some of them were already deeply analyzed for their prognostic capabilities in HCC (Chiang et al, 2008). *GGA3* and *ACACA*, implicated in regulating intracellular trafficking and in the fatty acid synthesis, respectively, are known to be involved in HCC pathogenesis and associated with a poor prognosis in HCC (Hu et al, 2015; Jiang et al, 2015). On the other hand, *SRRM2* and *LUC7L3* are involved in splicing processes and were previously reported to be associated with a poor prognosis in non-hepatocellular cancers (Malouf et al, 2014; Tomsic et al, 2015). We found that *LUC7L3* and *SRRM2* were heterogeneously expressed among the HCC clusters and their overexpression was almost confined in the clusters displaying a more proliferating and instable gene expression profile suggesting a role in HCC prognosis and progression. We next aimed to fine-map the intratumor relationship between HBV expression and host gene expression. We performed a lineage reconstruction on Smart-Seq2 data and observed again a single trajectory connecting cluster 1 (PHH), clusters 2/3, and finally cluster 4. Only cluster 5 did not show strong links with any other cluster, and therefore appeared disconnected (Fig 4A). Whereas all HCC cells were HBV-RNA positive (Fig 4B), an intratumoral, marked difference in HBV RNA levels was detected. HBV-RNA levels significantly differed between the clusters (Fig 4C), with the minimum HBV RNA level in cluster 4 which also expressed poor prognosis markers (Fig 3D). We used SOM of pseudo-temporal expression profiles to inspect the changes of known HBV host factors along the intratumor pseudo-differentiation trajectory (Fig 4D). Interestingly, several HBV replication enhancers, for example, *HLF* and *JUN* (Fig 4D) appeared to be up-regulated in the PHH and the most differentiated HCC cells of the same tumor. This is in line with the previously observed results on the intertumoral level (Fig 2). On the other hand, several HBV inhibitors, for example, *ONECUT1*, *RFX1*, and *SIRT1* (Turton et al, 2020) (Fig 4D) appeared to be up-regulated in less differentiated cancer cells. GSEA within the HBV-RNA–containing module 19 using the GSEA online "Investigate Gene Sets" tool revealed a significant enrichment of 16 genes (FDR = $2.22 \times 10^{-6}$) associated with liver specific genes (Hsiao et al, 2001), and of 18 genes (FDR = $1.54 \times 10^{-7}$) associated with a more differentiated liver cancer phenotype according the Hoshida "subtype S3" signature (Hoshida et al, 2009) (see Table S6). Both sets include Glucose transporter 2-encoding *SLC2A2*, highly expressed in hepatocytes and favorable prognostic marker in HCC (Uhlén et al, 2015; Uhlen et al, 2017), confirming intratumorally the association between HBV RNA levels and cell differentiation as we previously described at inter-tumor level (Fig 2). Analysis of HBV reads in single cell expression along the differentiation trajectory confirmed the hypothesis that HBV reads are higher in more differentiated cells also

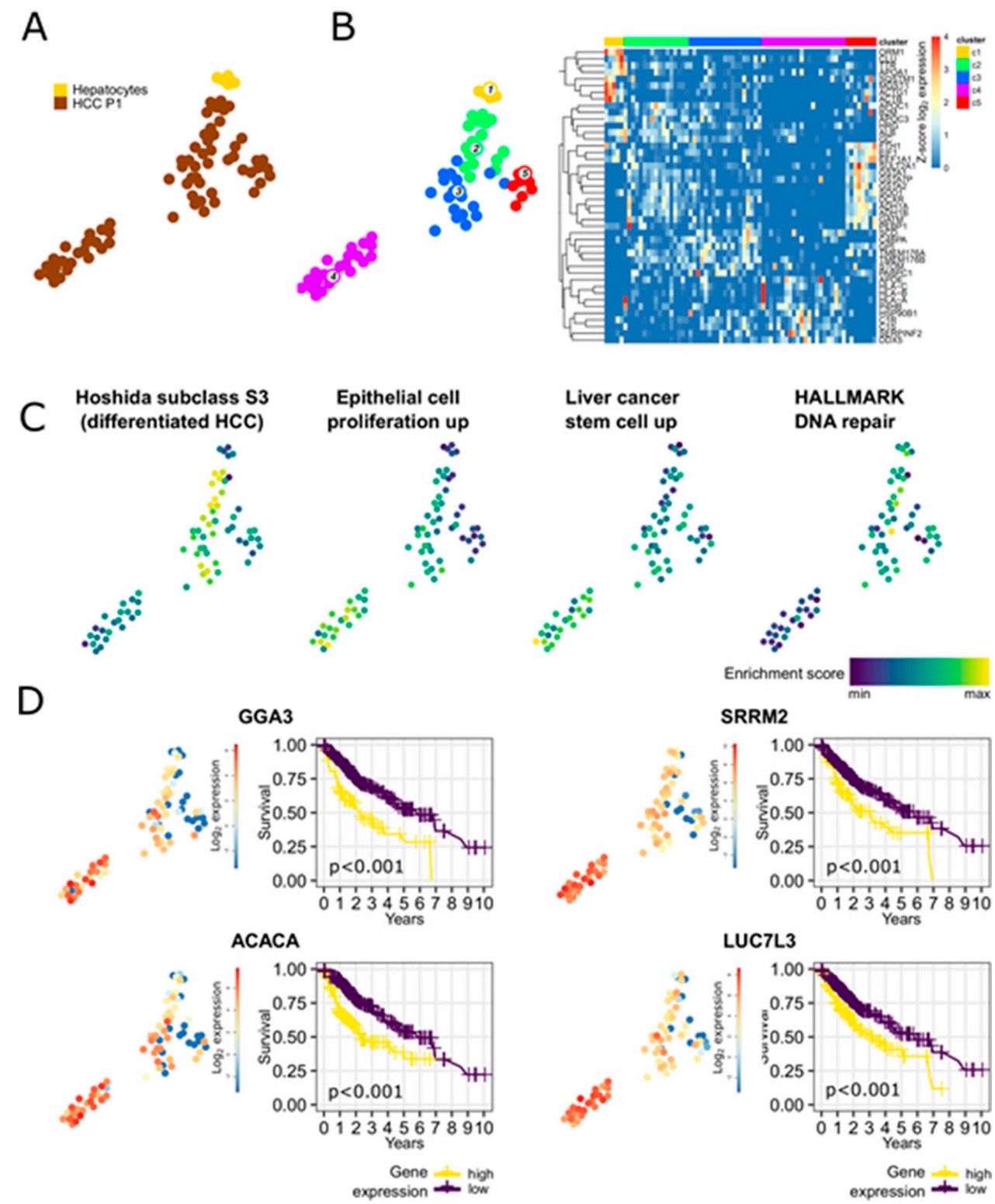

**Figure 3. Intratumor heterogeneity of HCC P1 analyzed by Smart-Seq2, a full-length scRNA-seq.**
**(A)** Fruchterman Reingold map of HCC P1 and healthy hepatocytes sequenced by Smart-Seq2. **(B)** Unsupervised k-medoids clustering of all single cells resulted in five clusters (left) and most differentially expressed genes throughout them (heat map, right). **(C)** Single-cell pathway enrichment analysis showing heterogeneity in enrichment for pathways associated with tumor differentiation (Hoshida subclass S3 and DNA repair) or undifferentiation/proliferation (Epithelial cell proliferation and Liver cancer stem cell) with a more aggressive and undifferentiated phenotype in cluster 4. **(D)** Poor prognosis genes in liver cancer are higher expressed in cluster 4. Kaplan–Meier curves of survival data of liver cancer patients from TCGA data.

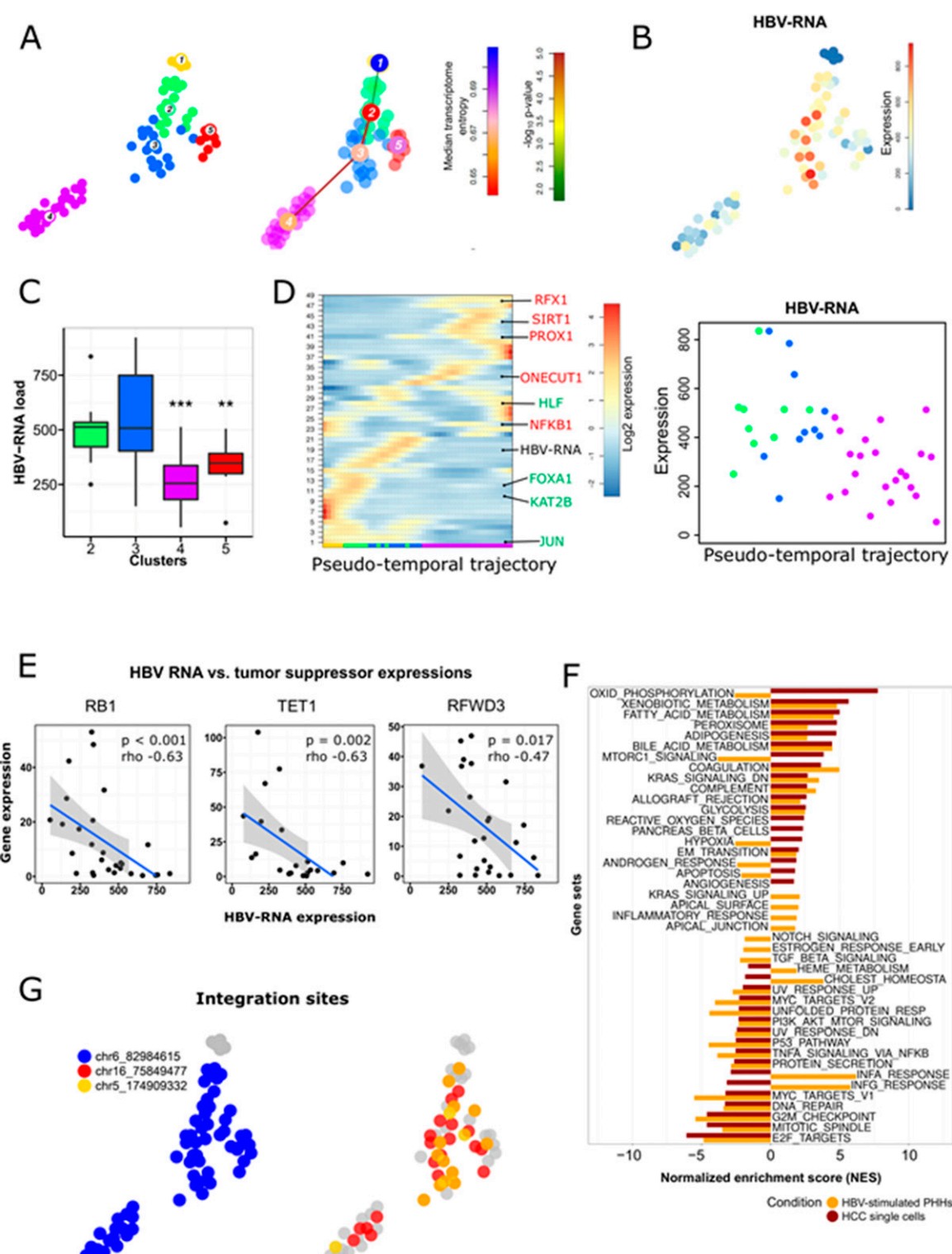

**Figure 4. Intratumor heterogeneity of HBV-RNA in HCC P1.**
**(A)** *t*-SNE map of HCC P1 cells and healthy hepatocytes sequenced using Smart-Seq2 (left), and the results of a differentiation lineage reconstruction by pseudo-temporal ordering the single cell's expression profiles. The calculated trajectory, starting in hepatocytes (cluster 1), is progressively following the differentiation status of cells (see Fig 3C), and ends in less differentiated cancer cells (cluster 4). **(B)** Expression t-SNE map showing higher HBV-RNA in the more differentiated clusters (2 and 3). **(C)** Viral compartmentalization with significant differences (Mann–Whitney test. *P < 0.05; **P < 0.01, ***P < 0.001) of HBV reads in HCC cell clusters. Colors indicating cell clusters refer to Fig 3B. **(D)** Self-organizing map of pseudo-temporal expression profiles along the differentiation branch; HBV-RNA and representative genes enhancing (green) or inhibiting (red) HBV replication are shown in the corresponding modules. **(A)** Colors indicating cell clusters along the trajectory refer to (A). **(E)** HBV-RNA level

at intratumor level (Fig 4D). Collectively, these findings confirm a compartmentalization of HBV which is most likely due to different host factor expression dependent on cellular differentiation (Fig 2) as shown above.

To identify functionally relevant virus-host interactions for hepatocarcinogenesis, we analyzed the correlation between HBV reads and expression of oncogenes and TSGs from Cancer Gene Census database (https://cancer.sanger.ac.uk/census, Table S7). HBV reads in single cells correlated negatively with *RB1*, encoding the tumor suppressor retinoblastoma 1, and *TET1*, encoding ten-eleven translocation methylcytosine dioxygenase 1, known for their role in HCC development (Tao et al, 2015; Kent et al, 2017) (Fig 4E). We also identified unknown TSG implicated in HCC pathogenesis and negatively correlating with HBV-RNA, for example, *RFWD3*, encoding a E3 ubiquitin ligase stabilizing P53 expression (Fu et al, 2010), which has been already described as TSG in non-small cell lung cancer and hematological malignancies (Fig 4E). Interestingly, no known tumor oncogene strongly correlated (Rho > 0.4) with HBV-RNA levels. Although this aspect of the study is only descriptive, these observations may suggest that HBV modulation of TSG, rather than oncogene, would be more relevant for HBV-induced carcinogenesis in our patients. Further functional analyses would be required to determine a putative direct action of the virus and its proteins on the expression of this TSG. To understand the functional impact of HBV on HCC gene expression, we performed a GSEA pathway analysis using the MSigDB Hallmark collection and identified cellular pathways enriched for genes correlating with HBV reads (Fig 4F). Notably, genes positively correlating with high HBV levels were associated with an induction of bile acid- and fatty acid metabolism—a hallmark of differentiated hepatocytes. These data again confirming the association of HBV replication and cancer cell differentiation. This result is in line with the recent observation that the presence of pgRNA in HCC is associated with low expression of cell cycle- and DNA repair–related genes (Halgand et al, 2018). We observed a similar down-regulation of mitotic spindle and DNA repair pathways at the single cell level (Fig 4F). Furthermore, the perturbation of gene expression in cancer cells harboring HBV transcripts was highly similar to the gene expression profile of HBV-infected PHH (Yoneda et al, 2016). A down-regulation of E2F- and MYC targets was observed, as well as a strong induction of xenobiotic metabolism and adipogenesis (Fig 4F). Although the effectors of gene expression profiles may be different in HCC cells and PHH, this observation suggests that HBV infection induces long-term modifications of the hepatocyte transcriptomic profile, which in turn is likely to be involved in the development of liver disease and carcinogenesis. Interestingly, several cancer-related pathways were only up-regulated in HCC-derived single cells containing HBV reads, suggesting a specific impact of viral gene expression on cancer cell gene expression. This is illustrated for example, by the specific up-regulation of hypoxia gene set expression in HCC cells with positive HBV reads (Fig 4F). Hypoxia plays a key role in hepatocarcinogenesis and liver tumor progression, through the ability of the hypoxia inducible factor 1 $\alpha$ to target the expression of oncogenic genes such as the proliferation-specific transcription factor Forkhead

box M1 (Lin & Wu, 2015). Hypoxia-inducible factor 1 $\alpha$ overexpression in HCC has been correlated with worse clinical outcomes and is considered as a poor prognosis factor and molecular target for liver disease therapy (Lin & Wu, 2015; Ju et al, 2016).

As the integration of the HBV genome into host cell has been suggested as an important mechanism of HBV-induced liver carcinogenesis (Levrero & Zucman-Rossi, 2016), we investigated whether HBV integration was detectable at the single-cell level through Smart-Seq2. To address this question, we analyzed discordant pairs of reads where read mates were mapping to both the HBV and the human genome, as allowed only by full transcript single cell sequencing protocols like Smart-Seq2. Whereas reads link HBV with unannotated regions of the human genome, we did not detect any integration within the coding sequence of key carcinogenesis-related genes suggesting that HBV integration was not the primary carcinogenetic mechanism in the sequenced HCC. We observed that at least three integration sites on chromosomes 5, 6, and 16 were shared by cells coming from the four different clusters, revealing that the analyzed HCC cells have a unique clonal origin (Fig 4G). Therefore, the HBV integration background is most likely to be similar in each cell, suggesting that the observed differences in HBV-RNA levels may be due to virus replication (Fig 4).

## Discussion

HBV infection is a leading cause of HCC worldwide. To date, the molecular mechanisms of viral hepatocarcinogenesis are poorly understood. In this study, using an innovative approach at the single cell level, we demonstrated a robust correlation between HBV reads and cancer differentiation. Importantly, we detected an active HBV replication within the highly differentiated tumor tissue, correlating with the expression of HBV host factors. In addition, we unrevealed novel pathways mediating HBV-induced carcinogenesis as well as novel putative integration sites for HBV. The clinical implications of this study are twofold: despite low levels of HBV-replication in these patients, HBV reads were associated with perturbation of expression of genes involved in carcinogenesis, mainly TSG. Furthermore, HBV reads correlated with cell differentiation at intertumoral-, intratumor, and single cell levels. Our results showed for the first time on single cell level that HBV replicates within hepatic tumors and that HBV replication is associated with the expression of HBV-related host factors crucial for the replication of the virus. Our data suggest that even at low expression levels, HBV perturbs the expression of cancer-related genes. This underlines the importance of curing HBV chronic infection and highlights the limitation of current standard of care suppressing but rarely eliminating HBV infection. It is of interest to note that viral compartmentalization was also suggested for HCV (Hedegaard et al, 2017). Whether there are shared molecular mechanisms for HBV and HCV compartmentalization remains to be determined. One limitation of our study is the small number of patient samples. However, the reproducibility of key findings within

---

inversely correlated with tumor suppressor genes *RB1*, *TET1*, and *RFWD3* at single-cell level; *P*-values for Spearman correlations. **(F)** Pathway analysis comparing enriched pathways in HBV-stimulated PHHs (GEO ID GSE69590), and HCC P1 single cells using, as gene set, genes significantly correlating with HBV RNA level. **(G)** Top three HBV integrations sites per single cell showing clonal origin of HCC P1.

and among the tumors and by the two complementary RNA-Seq protocols SMART-seq2 (providing full mRNA coverage for integration site detection) and mCEL-seq2 (providing deep coverage also of lowly expressed host factors) suggest that our findings are relevant for viral hepatocarcinogenesis in general. In conclusion, we show that the virus is compartmentalized within tumors dependent on HBV host factor expression. Analysis of HBV-host cell interactions at the single cell levels revealed previously undiscovered driver candidate pathways for carcinogenesis providing opportunities for novel approaches for prevention and treatment of virus-induced HCC. The perturbation of pro-carcinogenic gene expression even at low HBV levels highlights the need of HBV cure to eliminate HCC risk.

# Materials and Methods

### Human subjects

Human liver tissue samples were obtained from patients who had undergone liver resections between 2014 and 2019 at the Center for Digestive and Liver Disease (Pôle Hépato-digestif) at the Strasbourg University Hospitals, University of Strasbourg, France. All patients provided written informed consent. The protocols followed the ethical principles of the declaration of Helsinki and were approved by the local Ethics Committee of the University of Strasbourg Hospitals and by the French Ministry of Education and Research (CPP 10-17, Ministère de l'Education Nationale, de l'Enseignement Supérieur et de la Recherche; approval number DC-2016-2616). Data protection was performed according to the EU legislation regarding privacy and confidentiality during personal data collection and processing (Directive 95/46/EC of the European Parliament and of the Council of the 24 October 1995). PHHs were obtained from liver tissue from patients undergoing liver resection for liver metastasis at the Strasbourg University Hospitals with informed consent as described (Krieger et al, 2010). Protocols were approved by the local Ethics Committee of the Strasbourg University Hospitals (CPP) and the Ministry of Higher Education and Research of France (DC 2016 2616).

### Single-cell isolation and miniaturized mCEL-seq2 sequencing

Single cells were isolated from HCC tissue using collagenase digestion as previously described (Krieger et al, 2010; Aizarani et al, 2019) and sorted on an SH800 cell sorter (Sony) using a 100-µm chip. Zombie Green (BioLegend) was used as a viability dye and cells were stained with human-specific antibodies against CD45 (Cat. no. 304023; BioLegend) and PECAM1 (Cat. no. 303111; BioLegend). Viable cells were sorted in an unbiased fashion or from specific populations based on the expression of markers into the wells of 384-well plates containing lysis buffer. Cancer epithelial cells were enriched from the CD45-/ PECAM-population. Single-cell RNA sequencing was performed according to a modified CEL-Seq2 protocol as described (Muraro et al, 2016). Reads were aligned to the human hG19 UCSC reference as well as the HBV genome (included as an additional chromosome) using hisat2 and suppressing discordant alignments for paired reads. Dataset were filtered, analyzed, and normalized using RaceID3 (Aizarani et al,

2019). As quality control, we removed all the cells with more than 80% of mitochondrial genes. Transcripts correlating to KCNQ1OT1 with a Pearson's correlation coefficient of >0.4 were removed. RaceID3 was run with the following parameters: mintotal = 1,500, minexpr = 2, minnumber = 5, knn = 10. This step filtered out the cells with <1,500 transcripts found at least in number of two in a minimum of five cells. A total of 486 cells for HCC P1 and 452 from HCC P2 were then analyzed. For normalization, the total transcript counts in each cell were normalized to 1 and multiplied by the minimum total transcript count across all cells that passed the quality control threshold. Integration of 200 random hepatocytes from the human cell atlas (Aizarani et al, 2019) were performed using RaceID3 with the same parameters as mentioned before, applying RaceID batch effect and filtering out HBV-RNA from minimum transcript count and clustering.

### Single-cell isolation and Smart-seq2 sequencing

For deep intratumor analysis through Smart-seq2, single cells were isolated and sorted into 96-well plates using MoFlo sorter (Beckman Coulters). Human single hepatocytes were isolated from healthy liver tissue as described (Krieger et al, 2010). Tumor dissociation was performed using gentleMACS dissociator (Milteny) and the Tumor Dissociation kit for human biopsies (130-095-929; Milteny) following the manufacturer's procedure. Single cells were isolated and sorted into 96-well plates using a MoFlo sorter (Beckman Coulters). Single cells were lysed in 15 µl TCL buffer (QIAGEN) supplemented with 1% 2-mercaptoethanol. Cellular mRNA was isolated and analyzed as described (Shalek et al, 2014; Trombetta et al, 2014). Paired-end 25-bp reads were sequenced for control PHHs and HCC cells (one plate of 96 single cells for each) using the Smart-Seq2 protocol (Picelli et al, 2013; Picelli et al, 2014; Trombetta et al, 2014). Reads were aligned to the human hG19 UCSC reference as well as the HBV (strain ayw) reference genome (GenBank: V01460.1, included as an additional chromosome) using hisat2 and suppressing discordant alignments for paired reads. Reads were counted using htseq-count (Anders et al, 2015). Following steps were performed in R using RaceID3 (Masuda et al, 2019; Grün, 2020; R-Core-Team, 2020). Cells expressing >2% of KCNQ1OT1 transcripts, a marker of low-quality cells (Grün et al, 2016), were removed. Transcripts correlating to KCNQ1OT1 with a Pearson's correlation coefficient of >0.4 were also removed. RaceID3 was run with the following parameters: mintotal = 1,000, minexpr = 2, minnumber = 5, knn = 10. This step filtered out the cells with <1,000 transcripts found at least in number of two in a minimum of five cells. HBV-RNA was filtered out for minimum transcript count and clustering. Imputing of gene expression using five nearest neighbors was performed. The filtered read count matrix was normalized, and subsequent analyses were performed based on it. This resulted in a first set of 75 cells from the HCC. Using known hepatocyte, HCC, and epithelial-specific markers such as *ALB*, *GPC3*, and *KRT8*, 66 cancer epithelial cells were selected from HCC P1 and integrated with five PHHs. The remaining nine cells of HCC P1 were annotated as non-parenchymal cells.

### Single-cell clustering and lineage tree analysis

Single cell clustering, marker gene analysis, lineage tree calculations, and the calculation of SOMs of modified CEL-seq2 and

Smart-seq2 data were performed using the RaceID/StemID pipeline as described before (Aizarani et al, 2019; Grün, 2020) using unfiltered normalized dataset and with the following parameters: corthr = 0.75, minsom = 8 for mCEL-Seq2 data and corthr = 0.85, minsom = 8 for Smart-seq2 data.

### Quantification of HBV cccDNA and pgRNA in tumor samples

Total DNA from both tumor samples and from healthy primary human hepatocytes from a HBV-negative patient was extracted as previously described (Verrier et al, 2016). cccDNA levels were then quantified by qPCR as described (Verrier et al, 2016). Alternatively, Total RNA from both tumor samples and from healthy primary human hepatocytes from a HBV-negative patient was extracted as previously described (Verrier et al, 2016). pgRNA levels were then quantified by RT-qPCR as described (Verrier et al, 2016).

### HBV integration sites

Reads were mapped again using hisat2 as described above, but with the allowance for paired reads to be mapped to different chromosomes. Multiple discordant pairs of reads mapping uniquely to both, a human gene and the HBV chromosome, were considered as integration events in a single cell.

### GSEA of single cells

The filtered and normalized read count matrix served as input for performing single sample gene set enrichments for each cell applying the R package gene set variation analysis (Hänzelmann et al, 2013; R-Core-Team, 2020). Enrichments were calculated for MSigDB v7.0 (Subramanian et al, 2005) pathways applying the "ssgsea" method with options "min.sz = 5," "max.sz = 12,000," and "ssgsea.norm = T." Enrichment scores were plotted using the ggplot2 package in R (Wickham, 2016b).

### GSEA of clusters in association with HBV loads

Correlations between individual gene expressions and HBV loads for all annotated human genes were determined through the Spearman's rank correlation significance (P-value), and genes were then ranked according to the Spearman's P-value. Enrichment analysis for MSigDB v7.0 pathways was assessed using the pre-ranked GSEA (Subramanian et al, 2005) according to the obtained ranking. FDR below 0.05 was considered as statistically significant. Another experimental dataset (GSE69590) was used in this study, where gene expression of HBV-stimulated PHHs was compared to naïve PHHs. Pathway analysis was also assessed through the pre-ranked GSEA based on P-values of DESeq2 differential expression analysis. The normalized enrichment scores of pathways significantly differentially expressed between HBV-stimulated PHHs and naïve PHHs were then compared to the normalized enrichment score of pathways significantly modulated in association with HBV load in single cells.

### GSEA of SOM modules

We used the online available "Investigate Gene Sets" from https://www.gsea-msigdb.org to calculate gene set enrichments in SOM modules. Results were downloaded as tsv files.

### Survival analysis

Prognostic genes information for liver cancer was imported from The Human Protein Atlas (Uhlén et al, 2015). The survival data for specific gene was retrieved from the TCGA website (Cancer Genome Atlas Research Network et al, 2013), and survival curves were calculated and drawn using corresponding R packages (survminer, survival, and ggplot2) (Therneau & Grambsch, 2000; Wickham, 2016a, 2016b).

## Data Availability

The SRA study accession numbers for the data reported in this study are SRP165160 and SRP275756. The results shown here are in part based upon data generated by the TCGA Research Network: https://www.cancer.gov/tcga.

## Supplementary Information

## Acknowledgements

We thank Mauro Muraro (Single Cell Discoveries, Utrecht, Netherlands) for mCEL-seq2 sequencing. This work was supported by Inserm, the University of Strasbourg, the European Union (ERC-2014-AdG-671231-HEPCIR and Horizon 2020 research and innovation programme under grant agreement 667273—HEPCAR), the Agence Nationale de Recherches sur le Sida et les Hépatites Virales (ANRS ECTZ104527), the French Cancer Agency (TheraHCC 2.0 IHU201901299), and the US National Institutes of Health/National Institute of Allergy and Infectious Diseases/ National Cancer Institute (NIH/NIAID U19 AI123862-01, NIH/NIAID R03 AI131066, NIH/NCI R21 CA209940, R01CA233794). This work has been published under the framework of the LabEx ANR-10-LAB-28 and INSERM—Plan Cancer "single cell" 2018 (HCCMICTAR) and benefits from a funding from the state managed by the French National Research Agency as part of the Investments for the Future (Investissements d'Avenir) program. A Saviano acknowledges a fellowship co-funded by the Région Alsace, France, the LabEx HepSys, and the Institut Hospitalo-Universitaire de Strasbourg (IHU), France.

### Author Contributions

F Jühling: data curation, software, formal analysis, validation, investigation, methodology, and writing—original draft, review, and editing.
A Saviano: data curation, software, formal analysis, validation, investigation, methodology, and writing—original draft, review, and editing.
C Ponsolles: investigation and methodology.
L Heydmann: investigation and methodology.
E Crouchet: investigation and methodology.
SC Durand: investigation and methodology.
H El Saghire: data curation, software, and methodology.
E Felli: resources.
V Lindner: resources, data curation, and methodology.
P Pessaux: resources.

N Pochet: software and methodology.

C Schuster: validation, visualization, and methodology.

ER Verrier: conceptualization, supervision, funding acquisition, investigation, methodology, project administration, and writing—original draft, review, and editing.

TF Baumert: conceptualization, supervision, funding acquisition, investigation, methodology, project administration, and writing—original draft, review, and editing.

## Conflict of Interest Statement

The authors declare that they have no conflict of interest.

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
