## [Reviewer comments · Life Science Alliance]

Life Science Alliance

Hepatitis B virus compartmentalization and single cell differentiation in hepatocellular carcinoma

Frank Jühling, Antonio Saviano, Clara Ponsolles, Laura Heydmann, Emilie Crouchet, Sarah Durand, Houssein El Saghire, Emanuele Felli, Véronique Lindner, Patrick Pessaux, Nathalie Pochet, Catherine Schuster, Eloi Verrier, and Thomas Baumert

DOI: <https://doi.org/10.26508/lsa.202101036>

Corresponding author(s): Thomas Baumert, University of Strasbourg

Review Timeline:

Submission Date:	2021-01-25
Editorial Decision:	2021-03-24
Revision Received:	2021-06-22
Editorial Decision:	2021-06-22
Revision Received:	2021-07-05
Accepted:	2021-07-06

Transaction Report:

March 24, 2021

Re: Life Science Alliance manuscript #LSA-2021-01036-T

Thomas F. Baumert
University of Strasbourg
IVH - Inserm U1110
3 rue Koeberlé
Strasbourg 67000
France

Dear Dr. Baumert,

Thank you for submitting your manuscript entitled "Hepatitis B virus compartmentalization and single cell differentiation in hepatocellular carcinoma" to Life Science Alliance (LSA). The manuscript was assessed by expert reviewers, whose comments are appended to this letter.

We apologize for this unusual and extended delay in getting back to you. As you will note from the reviewers' comments below, Reviewer 1 is quite excited about these findings, but Reviewer 2 does have some concerns, with the main one being the assumption that HCC tumors progress in a monophyletic evolution along a single path - this concern should be addressed by further discussion both in the manuscript text and the pbp rebuttal. All the other concerns raised by the reviewers should be addressed as well. We, thus, encourage you to submit a revised version of the manuscript back to LSA that responds to all of the reviewers' points.

Thank you for this interesting contribution to Life Science Alliance. We are looking forward to receiving your revised manuscript.

Sincerely,

Shachi Bhatt, Ph.D.

Executive Editor

Life Science Alliance

<https://www.lsjournal.org/>

Interested in an editorial career? EMBO Solutions is hiring a Scientific Editor to join the international Life Science Alliance team. Find out more here -

https://www.embo.org/documents/jobs/Vacancy_Notice_Scientific_editor_LSA.pdf

B. MANUSCRIPT ORGANIZATION AND FORMATTING:

Reviewer #1 (Comments to the Authors (Required)):

This is a very original paper studying at the single cell level and in hepatocellular carcinoma cells, the

relationships between HBV expression and transcriptional regulations. The major finding was the correlation between expression of HBV RNA and the cellular differentiation. The second important issue is the heterogeneity of liver cancerous cells.

This is a major paper. Not the first using single cell RNA seq technology in the field of HBV infection, but certainly the most interesting one.

The results are clearly presented and the biostatistical part and figures are outstanding.

Only one small regret is the absence of data on HBV variability among different cells and according the cellular differentiation. This minor remark should not delay the publication of this remarkable paper.

Reviewer #2 (Comments to the Authors (Required)):

Reviewing report for Life Science Alliance of the manuscript entitled "Hepatitis B virus compartmentalization and single cell differentiation in hepatocellular carcinoma" by Frank Jühling

In this manuscript, the authors analyzed by single-cell sequencing gene expression in two hepatitis B virus-associated Hepatocellular Carcinomas surgically resected from two different patients. Both patients presented with relatively low HBV DNA loads. A tumor was scored as well-to-moderately differentiated (P1) while the other was moderately differentiated (P2). In total, the authors sequenced the contents of 938 cells with the mCelSeq2 and 66 additional cells from patient P1 with Smart-Seq2. The authors observed that tumor cell heterogeneity was much higher than the heterogeneity of non-tumor cells (lymphocytes, myeloid cells, fibroblasts). The authors observed for the first time at the single cell level that HBV replication and expression in tumor cells depend on their differentiation state.

Major issues

The principal criticism that can be made to this work concerns the association of patients P1 and P2 on the same t-SNE map representing a differentiation lineage reconstruction (2C) and a pseudo-temporal expression profile (2E). This association stems from the assumption that HCC progression is the result of a monophyletic evolution along a single path. Hence, the last decades have produced various types of molecular classifications of HCC that clearly show that these tumors did not follow a single route to reach their final and deadly stage. It is abundantly referred to this situation in the manuscript (Hoshida S3, Boyault G1/G5-6).

The fact that two tumors do not have the same differentiation stages at resection does not mean that the more differentiated is programmed to pass through the same stages that the second one. This presentation is artificial and somehow misleading the reader.

Observation of HBV expression in different metachronous tumors from the same patient will have been sounder. Reconstitution of a single temporality from two tumors from two different patients with different differentiation levels is, thus, a kind of self-fulfilling prophecy.

The fact that cluster 5 of HCC P1 cannot be included in a pseudo-temporal trajectory (4D) within the same patient is another illustration of the artificial character of this part.

What are the respective numbers of cells sequenced in P1 and P2? What was the proportion of tumor cells and that of the others cell clusters visible on t-SNE map 1C?

Figure 1C and 1D: Authors present the t-SNE of ALB and GPC3 but do not replace the gens on the heatmap. Please correct. Add a "3" to GPC

The authors apparently consider the issue of HBV RNA expression in tumor as a recent preoccupation in the field and cite a paper of Losic (2020) to illustrate this questioning. There are,

however, publications tackling that subject since 1979 (David Aden, Nature). Without going back so far, it would be fair to acknowledge this fact.

What was the correlation between both sequencing techniques for P1? Is there a metric for that? Is there a contradiction with the fact the RB1 expression is inversely correlated with those of HBV transcripts? Presence of RB1 is supposedly the hallmark of non-cycling well-differentiated cells. The authors indicate that E2F genes are also downregulated in case of HBV expression. Is this apparent contradiction resulting from irrelevant observations?

On figure 4G, we can observe that HBV DNA integration are present in all 5 clusters. Does it mean that they do not have any impact on tumor cell differentiation and possibly evolution?

Minor issues

Abstract: HLF is not so famous that everybody knows it. Please indicate that it is a transcription factor.

"undiscovered pathways mediating viral carcinogenesis: The abstract is short and some names of these new pathways will be welcomed.

There are 2 "unravel" and 2 "undiscovered" in the abstract. A more extensive vocabulary will be more pleasant for the reader.

Introduction: what is reference XX?

Results: please add in the text the age and sex of P1 to be homogenous with P2

"Collectively" page 7, bottom: what does it mean? "Collectively" "Selectively"

Point-by-point response - LSA-2021-01036-TR

General response to Editor and Reviewers,

We thank the Editor and the Reviewers for having carefully reviewed our manuscript and for their positive feedback and comments. We addressed all the concerns by 1) improving the figure presentation. 2) clarifying our results and conclusions within the text 3) updating the bibliography 4) specifying methods when necessary.

Reviewer #1 (Comments to the Authors (Required)):

This is a very original paper studying at the single cell level and in hepatocellular carcinoma cells, the relationships between HBV expression and transcriptional regulations. The major finding was the correlation between expression of HBV RNA and the cellular differentiation. The second important issue is the heterogeneity of liver cancerous cells. This is a major paper. Not the first using single cell RNA seq technology in the field of HBV infection, but certainly the most interesting one. The results are clearly presented and the biostatistical part and figures are outstanding. Only one small regret is the absence of data on HBV variability among different cells and according to the cellular differentiation. This minor remark should not delay the publication of this remarkable paper.

We thank the reviewer for his/her very positive appreciation of our manuscript. According to this/her comments, we have further clarified the impact of study and results. Indeed, we described HBV variability among cells when correlating HBV RNA- and gene expression levels within single cells (Figure 2E, I, J, K and Figure 4D, E). Therefore, we have more clearly presented this analysis and discussed HBV variability according to cellular differentiation in the manuscript:

Line 180: “Since viral RNA expression correlated with tumor differentiation, we further characterized the phenotype of the HBV expressing cells using the survival data gene sets from TCGA. We found that SORD, encoding sorbitol dehydrogenase, and highly expressed in differentiated hepatocytes correlated with HBV expression (Uhlen et al., 2015) (Figure 2J)”.

The following sentence was added to the discussion:

Line 318: “Furthermore, HBV reads correlated with cell differentiation at intertumoral-, intratumor, and single cell level.”

Reviewer #2 (Comments to the Authors (Required)):

Major issues

The principal criticism that can be made to this work concerns the association of patients P1 and P2 on the same t-SNE map representing a differentiation lineage reconstruction (2C) and a

pseudo-temporal expression profile (2E). This association stems from the assumption that HCC progression is the result of a monophyletic evolution along a single path. Hence, the last decades have produced various types of molecular classifications of HCC that clearly show that these tumors did not follow a single route to reach their final and deadly stage. It is abundantly referred to this situation in the manuscript (Hoshida S3, Boyault G1/G5-6). The fact that two tumors do not have the same differentiation stages at resection does not mean that the more differentiated is programmed to pass through the same stages that the second one. This presentation is artificial and somehow misleading the reader. Observation of HBV expression in different metachronous tumors from the same patient will have been sounder. Reconstitution of a single temporality from two tumors from two different patients with different differentiation levels is, thus, a kind of self-fulfilling prophecy. The fact that cluster 5 of HCC P1 cannot be included in a pseudo-temporal trajectory (4D) within the same patient is another illustration of the artificial character of this part.

We thank the reviewer for this relevant comment. We agree that HCC development and progression cannot be summarized by a single evolution path and that the lineage reconstitution of the two tumors within a single t-SNE map is artificial. However, representing cells from different patients on a single t-SNE map is a classical illustration used in single cell studies, including liver studies such as (Aizarani, Saviano et al., 2019, Losic, Craig et al., 2020, MacParland, Liu et al., 2018), to highlight similarities and differences in a simple way.

In our study, we do not assess HCC phylogeny, but we used a fate analysis only to unbiasedly distinguish the differentiation grading of the two tumors and cell clusters without claiming that one tumor derives from the other or that there is only one evolution path. Pseudo-temporal expression profile is used here to hierarchize different cell clusters according to their similarity.

The text has been modified to clarify that point:

Line 111: **“To illustrate differences and similarities among single cell between the two patients, we artificially plotted Single-cell gene expressions on T-distributed Stochastic Neighbor Embedding (t-SNE) maps indicating cell similarities as is it commonly in single-cell studies, including liver ones (Aizarani, Saviano et al., 2019, Losic, Craig et al., 2020, MacParland, Liu et al., 2018)”**

Line 131: **“Although this approach does not imply an HCC monophyletic evolution from healthy hepatocytes to P2 HCC cells, it allowed to infer unbiased cell differentiation.”**

Line 137: **“We then evaluated pseudo-temporal expression profiles along the differentiation branch to identify differentially expressed between the different clusters.”**

What are the respective numbers of cells sequenced in P1 and P2? What was the proportion of tumor cells and that of the others cell clusters visible on t-SNE map 1C?

We agree with the reviewer that this information should be more clearly presented. The number of cells sequenced in P1 and P2 is now presented in Supplementary Table 2. The following sentence was added to the manuscript.

Line 110: **“A summary of the number of sequenced cells is presented in Supplementary Table 2.”**

We sequenced 486 cells from P1 and 452 cells from P2 and we provide here the proportion of the cell types. It is important to note that we sequenced cells using FACS selection to enrich for cancer cells while excluding mainly lymphocytes and endothelial cells via known cell markers (Aizarani et al. *Nature* 2019). Moreover, we did not present these data and explore tumor microenvironment since this was not the aim of study.

Figure 1C and 1D: Authors present the t-SNE of ALB and GPC3 but do not replace the gens on the heatmap. Please correct. Add a "3" to GPC

We thank the reviewer for the relevant comment: GPC3 was corrected in the Figure. Moreover, ALB was replaced in the heatmap. However, GPC3 was not among the marker genes characterizing any of the clusters and therefore cannot be placed in the heatmap. GPC3 was presented in Figure 1D as a classical and well described HCC marker.

The authors apparently consider the issue of HBV RNA expression in tumor as a recent preoccupation in the field and cite a paper of Losic (2020) to illustrate this questioning. There are, however, publications tackling that subject since 1979 (David Aden, *Nature*). Without going back so far, it would be fair to acknowledge this fact.

We thank the reviewer for having pointed this out. We agree what our presentation was confusing regarding this point. One sentence has been added to the introduction to clarify it.

Line 57: **“The question of HBV replication within tumor or tumor cells has been studied for more than 50 years (Aden, Fogel et al., 1979)”**

What was the correlation between both sequencing techniques for P1? Is there a metric for that?

We calculated Spearman’s correlation between both sequencing techniques by comparing all cancer cell clusters from Fig.1B and Fig.3B as indicated in the following figure (CEL-seq2 clusters on x-axis and Smart-Seq2 clusters on y axis). Clusters from both techniques and originating from patient P1 cluster together (see maximum numbers highlighted in green).

We think that this comparison is quite artificial because of the fundamental differences in the sequencing techniques, i.e., CEL-seq2 measuring the number of mRNA molecules according to similar 3’-ends, and Smart-seq2 covering full transcripts and thereby producing length-

dependent gene counts. For this reason, we decided not to show these results in our manuscript. However, we think that the aim why we applied deeper analysis using Smart-seq2 is clearly justified in the manuscript (Line 207).

Is there a contradiction with the fact the RB1 expression is inversely correlated with those of HBV transcripts? Presence of RB1 is supposedly the hallmark of non-cycling well-differentiated cells. The authors indicate that E2F genes are also downregulated in case of HBV expression. Is this apparent contradiction resulting from irrelevant observations?

We thank the reviewer for this comment. At this stage, we can only describe correlations between the expression levels of HBV and cellular factors. Although it appears counterintuitive that these genes are associated HBV transcripts, we may suggest that HBV itself is responsible for their downregulation, as it has been already described a modulation of the RB1 pathways, notably by loss of RB1 expression in HBV-induced HCC (Edamoto, Hara et al., 2003). downregulates. Further analysis would be required to determine a putative direct action of the virus of these genes.

The text was modified accordingly:

Line 273: “Although this aspect of the study is only descriptive, these observations may suggest that HBV modulation of TSG, rather than oncogene, would be more relevant for HBV-induced carcinogenesis in our patients. Further functional analyses would be required to determine a putative direct action of the virus and its proteins on the expression of this TSG.”

On figure 4G, we can observe that HBV DNA integration are present in all 5 clusters. Does it mean that they do not have any impact on tumor cell differentiation and possibly evolution?

We thank the reviewer for this comment. HBV DNA integration occurs very rapidly following viral infection (Tu, Budzinska et al., 2018). New integration events can then result from new infection events (Tu, Zhang et al., 2021). Multiple HBV integration events may then coexist in the same cell. The impact of HBV integration of cell differentiation will depend on the site of integration. In this context, HBV is not likely to exhibit specific integration sites within the genome as it is the case for retroviruses (Tu et al., 2021). In our study, we indeed observed that one integration site is shared by all the cell clusters, suggesting that all cells com from the same initial clone. It is likely that this integration site on chromosome 6 out of any transcriptionally active region has any impact on cellular activity. We highlighted in his section a proof-of-concept that SmartSeq2 is a suitable method for the study of HBV integration sites.

Minor issues

Abstract: HLF is not so famous that everybody knows it. Please indicate that it is a transcription factor.

We thank the reviewer for this comment. The mention that HLF is a transcription factor is already present in the manuscript, Line 182:

“In line with this result, we observed a significant positive correlation between HBV-RNA levels and the expression of HLF, a well described liver-specific transcription factor crucial for HBV replication (Turton et al., 2020)”

"undiscovered pathways mediating viral carcinogenesis: The abstract is short, and some names of these new pathways will be welcomed.

We thank the reviewer for this suggestion. One sentence has been added to the summary:

Line 29: “Analyses of virus-induced host responses identified previously undiscovered pathways mediating viral carcinogenesis, such as E2F- and MYC targets as well as adipogenesis.”

There are 2 "unravel" and 2 "undiscovered" in the abstract. A more extensive vocabulary will be more pleasant for the reader.

We agree with the reviewer and modify the abstract accordingly.

Line 31: “Mapping of fused HBV-host cell transcripts allowed the characterization of [...]”

Line 33: “[...] virus and cancer identifying new candidate pathways [...]”.

Introduction: what is reference XX?

We thank the reviewer for having pointed out this typo. It has been removed.

Results: please add in the text the age and sex of P1 to be homogenous with P2

We agree with the reviewer. This was corrected according, Line 73: **“Patient P1 was 61-year-old with normal liver function tests.”**

"Collectively" page 7, bottom: what does it mean? "Collectively" "Selectively"

We thank the reviewer. The typo has been corrected. Line 186: **“Collectively”.**

References

Aizarani N, Saviano A, Sagar, Mailly L, Durand S, Herman JS, Pessaux P, Baumert TF, Grun D (2019) A human liver cell atlas reveals heterogeneity and epithelial progenitors. *Nature* 572: 199-204

Edamoto Y, Hara A, Biernat W, Terracciano L, Cathomas G, Riehle HM, Matsuda M, Fujii H, Scoazec JY, Ohgaki H (2003) Alterations of RB1, p53 and Wnt pathways in hepatocellular

carcinomas associated with hepatitis C, hepatitis B and alcoholic liver cirrhosis. *International journal of cancer* 106: 334-41

Losic B, Craig AJ, Villacorta-Martin C, Martins-Filho SN, Akers N, Chen X, Ahsen ME, von Felden J, Labgaa I, D'Avola D, Allette K, Lira SA, Furtado GC, Garcia-Lezana T, Restrepo P, Stueck A, Ward SC, Fiel MI, Hiotis SP, Gunasekaran G et al. (2020) Intratumoral heterogeneity and clonal evolution in liver cancer. *Nature communications* 11: 291

MacParland SA, Liu JC, Ma XZ, Innes BT, Bartczak AM, Gage BK, Manuel J, Khuu N, Echeverri J, Linares I, Gupta R, Cheng ML, Liu LY, Camat D, Chung SW, Seliga RK, Shao Z, Lee E, Ogawa S, Ogawa M et al. (2018) Single cell RNA sequencing of human liver reveals distinct intrahepatic macrophage populations. *Nature communications* 9: 4383

Tu T, Budzinska MA, Vondran FWR, Shackel NA, Urban S (2018) Hepatitis B Virus DNA Integration Occurs Early in the Viral Life Cycle in an In Vitro Infection Model via Sodium Taurocholate Cotransporting Polypeptide-Dependent Uptake of Enveloped Virus Particles. *Journal of virology* 92

Tu T, Zhang H, Urban S (2021) Hepatitis B Virus DNA Integration: In Vitro Models for Investigating Viral Pathogenesis and Persistence. *Viruses* 13

June 22, 2021

RE: Life Science Alliance Manuscript #LSA-2021-01036-TR

Prof. Thomas F. Baumert
University of Strasbourg
IVH - Inserm U1110
3 rue Koeberlé
Strasbourg 67000
France

Dear Dr. Baumert,

Thank you for submitting your revised manuscript entitled "Hepatitis B virus compartmentalization and single cell differentiation in hepatocellular carcinoma". We would be happy to publish your paper in Life Science Alliance pending final revisions necessary to meet our formatting guidelines.

- please add ORCID ID for the corresponding author-you should have received instructions on how to do so
- please add Keywords and a Category for your manuscript in our system
- please add a Summary Blurb/Alternate Abstract in our system
- please use the [10 author names, et al.] format in your references (i.e. limit the author names to the first 10)
- please be sure to add callouts for the panels of all supplementary figures to your main manuscript text
- please add your table legends to the main manuscript text after the main and supplementary figure legends

Figure checks:

- Please replace the magnification levels with scale bars for Figure S1

A. FINAL FILES:

B. MANUSCRIPT ORGANIZATION AND FORMATTING:

Sincerely,

Eric Sawey, PhD
Executive Editor
Life Science Alliance

<http://www.lsajournal.org>

July 6, 2021

RE: Life Science Alliance Manuscript #LSA-2021-01036-TRR

Prof. Thomas F. Baumert
University of Strasbourg
IVH - Inserm U1110
3 rue Koeberlé
Strasbourg 67000
France

Dear Dr. Baumert,

Thank you for submitting your Research Article entitled "Hepatitis B virus compartmentalization and single cell differentiation in hepatocellular carcinoma". It is a pleasure to let you know that your manuscript is now accepted for publication in Life Science Alliance. Congratulations on this interesting work.

DISTRIBUTION OF MATERIALS:

Again, congratulations on a very nice paper. I hope you found the review process to be constructive and are pleased with how the manuscript was handled editorially. We look forward to future exciting submissions from your lab.

Sincerely,
